# Improved DDoS Detection Utilizing Deep Neural Networks and Feedforward Neural Networks as Autoencoder

**Ahmed Latif Yaser [1,2,*]**, **Hamdy M. Mousa [1]** and **Mahmoud Hussein [1]**

[1]  Computer Science Department, Faculty of Computers and Information, Menoufia University, Shebin Elkom 32511, Egypt

[2]  Department of Information Systems, College of Administration and Economics, University of Baghdad, Baghdad P.O. Box 10071, Iraq

*  Correspondence: ahmedlatif82@gmail.com

**Abstract:** Software-defined networking (SDN) is an innovative network paradigm, offering substantial control of network operation through a network's architecture. SDN is an ideal platform for implementing projects involving distributed applications, security solutions, and decentralized network administration in a multitenant data center environment due to its programmability. As its usage rapidly expands, network security threats are becoming more frequent, leading SDN security to be of significant concern. Machine-learning (ML) techniques for intrusion detection of DDoS attacks in SDN networks utilize standard datasets and fail to cover all classification aspects, resulting in under-coverage of attack diversity. This paper proposes a hybrid technique to recognize denial-of-service (DDoS) attacks that combine deep learning and feedforward neural networks as autoencoders. Two datasets were analyzed for the training and testing model, first statically and then iteratively. The auto-encoding model is constructed by stacking the input layer and hidden layer of self-encoding models' layer by layer, with each self-encoding model using a hidden layer. To evaluate our model, we use a three-part data split (train, test, and validate) rather than the common two-part split (train and test). The resulting proposed model achieved a higher accuracy for the static dataset, where for ISCX-IDS-2012 dataset, accuracy reached a high of 99.35% in training, 99.3% in validation and 99.99% in precision, recall, and F1-score. for the UNSW2018 dataset, the accuracy reached a high of 99.95% in training, 0.99.94% in validation, and 99.99% in precision, recall, and F1-score. In addition, the model achieved great results with a dynamic dataset (using an emulator), reaching a high of 97.68% in accuracy.

**Keywords:** autoencoder; denial-of-service (DDoS); deep neural network; DDoS detection; software-defined network (SDN)

## 1. Introduction

Software-defined networking (known as SDN) makes the management and programming of network systems easier. By separating the control and data planes, SDN increases network efficiency by putting everything in one place. Once a traditional network is configured with policies, it is difficult to change. Moreover, manually configuring a network is time-consuming and prone to mistakes, and it does not fully use the physical network infrastructure. SDN is widely used, solves these problems easily, and makes better use of network equipment.

In software-defined networking (SDN), the controller connects with the forwarding plane via a south-bound application programming interface (API) using a secure transport layer service. In this system, flow tables allow network switches to match traffic flows. When a packet reaches a switch, whose header areas do not match with the flow table, the packet is sent to the controller as a packet-in message. Then, the controller transmits a packet-out or flow-mod signal with specific flow rules, which is then integrated into the

flow table. This indicates that the next time a related packet comes to the switch, it can be acted upon without the need to refer to the controller [1,2].

While the high degree of centralization of the controllers greatly simplifies network administration, it is a security nightmare. If these controllers are subject to a large amount of requests, they can be taken offline, rendering the network inoperable [3]. This makes SDN highly vulnerable to distributed denial-of-service attacks (DDoS), which overload the capacity of the controller and the flow tables in the switch, halting the processing of packets and leading the network to be ineffective. This weakness necessitates the need for backup controllers.

DDoS attacks are one of the most common types of attacks. A DDoS attacker uses multiple compromised network devices to send numerous forged packets with random source IP addresses toward the target host in the same network, degrading service quality. By flooding the victim's device with these packets, the attacker attempts to deny legitimate users access to the services offered by the victim server.

However, backup controllers face similar challenges, and can also be attacked and taken offline if network traffic is directed at them [4]. Due to this possibility, a system for the early detection and mitigation of these attacks is required. A powerful intrusion detection system (IDS) would preserve network performance, increase data security, prevent the loss of intellectual property, and limit potential liability for compromised notes or network data [5]. This need has resulted in extensive research on DDoS detection techniques. Neural networks have emerged as one of the most widely used IDS tools [6].

Many contributions to research on DDoS detection models include ways to identify and quantify common characteristics of the massive sets of illegitimate traffic that are used to flood a victim's network during DDoS attacks. Our objective is to design a model that detects DDoS attacks using a hybrid technique for detecting malicious network flows using an autoencoder and deep neural networks. The proposed model prevents the overfitting of predetermined malicious patterns. The driving force behind this objective is the idea that the use of an autoencoder will develop a more accurate classifier model alongside the deep neural network model, similar to the traditional neural network model for detecting malicious network traffic. Our primary responsibilities are creating a data representation model utilizing autoencoder techniques and a malware flow detection model using a deep neural network. Experiments were conducted to evaluate the proposed solution. Finally, the results obtained are compared with other state-of-the-art techniques.

## 2. Related Works

In recent years, several method-based DNN algorithms have been developed. Nam [7] proposed two DDoS assault detection methods based on the self-organizing map. The proposed methods and their detection architecture utilize flexible and programmable SDN technology. The SDN controller enables us to execute sophisticated classification and detection algorithms rapidly. By creating a testbed environment, we successfully analyze the accuracy and computational requirements of our suggested methods. The experimental results demonstrate that these algorithms minimize processing time, while maintaining an acceptable level of precision.

Pekta and Acarman [8] proposed a model-based deep learning architecture that combines CNN and LSTM to learn spatial-temporal features of network flows. When tested on the ISCX 2012 dataset, the model achieved 99.09% in accuracy, 99.08% in recall, 99.10% in precision, and 99.09% in F1-score. For CICIDS2017, the model achieved 97.97%, 98.83%, 98.89%, and 98.86%, respectively. Elsayed et al. [9] provided a systematic benchmarking analysis of four existing machine-learning techniques for attack traffic detection in SDNs, SVM, J48m, Naive Bayes, and Random Forest. They identified the shortcomings of traditional machine-learning-based methods and laid the groundwork for a more robust framework. Their experiments used the NSL-KDD dataset, and their results showed that J48 achieved the best result compared with the three other ML techniques.

Sindian et al. [10] proposed an enhanced deep sparse autoencoder-based framework for detecting DDoS attacks, as well as a strategy for minimizing costs. The sparse autoencoder is used to extract datasets, and the SoftMax layer is used to determine whether traffic is malicious or not. Since intrusion detection methods occasionally produce wrong predictions, metrics, such as accuracy, precision, detection rate, and specificity, are used to evaluate the models. Their solution used the CICDoS2019 dataset [11], and successfully detected intrusions with high accuracy and a low false positive rate. The model achieved 98% in accuracy, a 98.1% detection rate, 91% in precision, and 98% in specificity. Radanliev et al. [12] provided a new mathematical approach for the integration of perception engine design concepts, edge computing, artificial intelligence, and machine learning to automate anomaly detection. This engine drives incremental change by applying artificial intelligence and machine learning embedded at the edge of the internet of things (IoT) network to provide secure, actionable, real-time intelligence for predictive cyber risk analytics. In their review, the authors reported that denial-of-service (DoS) and DDoS are the most common and dangerous IoT attacks, which can flood the network of IoT devices with traffic. These attacks lead to connection overload and network exhaustion, preventing IoT devices from communicating. The small computational power on high-end hardware makes it difficult to solve DDoS attacks. However, IoT aims to connect objects over the internet, and the SDN orchestrates the network management by decoupling the control and data planes. As a result, the SDN provides flexibility and programmability in the IoT network without disturbing the underlying architecture of existing implementations. Therefore, we limited the scope of our work to the detection of DDoS attacks in an SDN environment, and, as a result, this will serve IoT. Tang et al. [13] proposed a hybrid, unsupervised deep learning approach for detecting distributed denial-of-service (DDoS) attacks using a stack autoencoder and a one-class support vector machine (SAE-1SVM). The experimental results showed that with a small set of flow features, the proposed algorithm achieves an average accuracy of 99.35%. The SAE-1SVM demonstrates that it significantly reduces processing time, while maintaining a high detection rate. In conclusion, the SAE-1SVM detects anomalies in imbalanced and unlabeled datasets with high accuracy. Kushwah et al. [14] proposed a hybrid machine-learning-based technique to detect these attacks. The extreme learning machine (ELM) model and the black hole optimization algorithm implement the proposed technique. Several experiments proposed an evaluation of the performance of their proposed method. Additionally, several experiments were conducted using four benchmark datasets: NSL KDD, ISCX IDS 2012, CICIDS2017, and CICDDoS2019. With these four datasets, the accuracy reached 99.23%, 92.19%, 99.50%, and 99.80%, respectively. Moreover, a comparison is carried out on the following: Alternative ELM-based techniques, ANNs trained with blackhole optimization, backpropagation ANNs, and other state-of-the-art techniques. Gadze et al. [15] investigated deep-learning-based models for DDoS classification: Long short-term memory (LSTM) and convolutional neural networks (CNN). The dataset was dynamically generated via Mininet, using OpenFlow switches and Floodlight as an external controller. The results showed that RNN LSTM achieved an accuracy of 89.63%, outperforming linear-based models, such as SVM (86.85%) and Naive Bayes (82.61%). The KNN algorithm, a linear-based model, had an even higher accuracy than their model accuracy of 99.4%. Moreover, the model performed best when using a data split of 70/30 (train/test split ratios). Singha and Jang-Jaccard [16] proposed a hybrid autoencoder model called MSCNN-LSTM-AE, which uses a combination of a multiscale convolutional neural network (MSCNN) and LSTM to find anomalies in network traffic. The approach first uses the MSCNN-autoencoder to evaluate the spatial features of the dataset, then an LSTM-based autoencoder network is used to identify the temporal features of the latent space features learned from MSCNN-AE. For testing, the authors used UNSW-NB15 [17], NSL-KDD [18], and CICDDoS2019. Their model (MSCNN-LSTM-AE) achieved an accuracy of 93.76% and recall of 92.26%. Ivanova et al. [19] proposed an optimized feed-forward neural network model for detecting IoT-based DDoS attacks through network traffic analysis directed at a specific target, which could be monitored continuously

by a tap. The proposed model applies to DoS and DDoS attacks involving TCP, UDP, and HTTP flood, keylogging, data exfiltration, OS fingerprinting, and service scan activities. It simply distinguishes this network traffic from normal network flows. As a solver, the neural network employs Adam optimization and the hyperbolic tangent activation function in all neurons from a single hidden layer. Depending on the targeted accuracy and processing speed, the number of hidden neurons can be varied. Testing on the BotIoT dataset reveals that developed models can be used with 8 or 10 features and have a discrimination error of $4.91 \times 10^{-3}$%. Prasad et al. [20] proposed a multimode framework based on voting to combat volumetric DDoS (VMFCVD) attacks. VMFCVD is based on three different detection modes: Fast detection mode (FDM), defensive fast detection mode (DFDM), and high accuracy mode (HAM). FDM is designed to classify network traffic when a server is under attack. The highly dimensional and reduced dataset aids FDM's detection speed. In most cases, the dimension reduction for FDM was greater than 97%, while maintaining an accuracy of 99.9% during our experiment. DFDM is an enhanced version of FDM that improves the detection accuracy of malicious network traffic by tightening the detection technique. HAM focuses on detection accuracy, outperforming FDM and DFDM significantly. When the server is stable, HAM is activated. VMFCVD has been extensively tested on the most recent benchmark DDoS and botnet datasets, including the UNSW NB15, UNSW2018 BoTIoT, CSE-CIC-IDS2018 (BoT and DDoS), CICIDS2017 (BoT and DDoS), DoHBrw2020, NBaIoT2018 (Mirai), and CICDDoS2019 (DNS, LDAP, SSDP, and SYN). The results of VMFCVD show that it outperforms recent studies. When the server is under a DDoS attack, VMFCVD performs remarkably.

## 3. The Concept of Detection DDoS Attacks in SDN

Distributed denial-of-service (DDoS) attacks are generally carried out by several machines. These attacks follow a similar pattern to a basic denial-of-service (DoS) attack. However, the use of multiple machines simultaneously as separate origins of attack amplifies the attack's impact, while making it challenging to locate the attackers. The attacker forms a network of machines, consisting of a master (Master) and many remote hosts (Slaves). During the course of the attack, the attacker connects to the master, which sends an order to all remote hosts. Then, these hosts attack the target using a technique chosen by the attacker [21].

Defense mechanisms against DDoS attacks have become one of the most significant challenges in network security. Consequently, a large number of defense classifications and taxonomies have emerged. One important way to categorize defense options is through the main characteristics of their defense. This results in three main categories and policies: (1) Stopping attacks before they reach the target [22] with firewalls as an example of this prevention mechanism; (2) attacking detection through the identification of anomalies in the traffic entering the network; (3) identifying the attack's ultimate origin. This last technique is complicated by two aspects of the IP protocol [23]. First, it can be easy for an attacker to spoof source IP addresses. Second, one cannot know the full end-to-end path of a packet. SDN presents a solution to these shortcomings, thanks to its holistic view of the entire network. Additionally, SDN makes it possible to organize a set of OpenFlow switches through a single controller, allowing the centralization of the network control plane. This centralization makes it more viable to trace the end-to-end path of a packet, as the controller has a global vision of the network [24].

Deep-learning (DL) algorithms are used for threat detection, bandwidth optimization, power efficiency, and network traffic management. In machine learning, data are of the utmost importance for decision-making, as opposed to specific conditions presented by the algorithm [25]. DL algorithms are classified into three types: Supervised, unsupervised, and reinforcement learning. In supervised learning, labeled data are utilized for classification and regression. Unsupervised learning focuses on the classification of unlabeled data into distinct classes [26]. This work is focused on investigating the use of two common deep-learning techniques, LSTM and CNN, and their integration with an autoencoder.

## 4. Proposed Model Structure

The proposed DDoS detection model presents the implementation of SDN along with a method using deep learning to classify network traffic and construct a classification model. The proposed model includes a 2-hidden-layer autoencoder network with sigmoid activation functions, as shown in Figure 1. Model testing is carried out using the intrusion detection evaluation dataset ISCXIDS2012 [27], as recommended in [28]. During attack simulations, the controller routes the traffic entering the SDN platform by modifying the flow tables. By examining the flow table's rules, the controller can decide whether to forward, drop, or block traffic, employing machine-learning algorithms to determine the optimal routing path. The algorithm provides a knowledge base for decision-making when classifying new flow instances, taking information from previously known classes in the supervised learning portion.

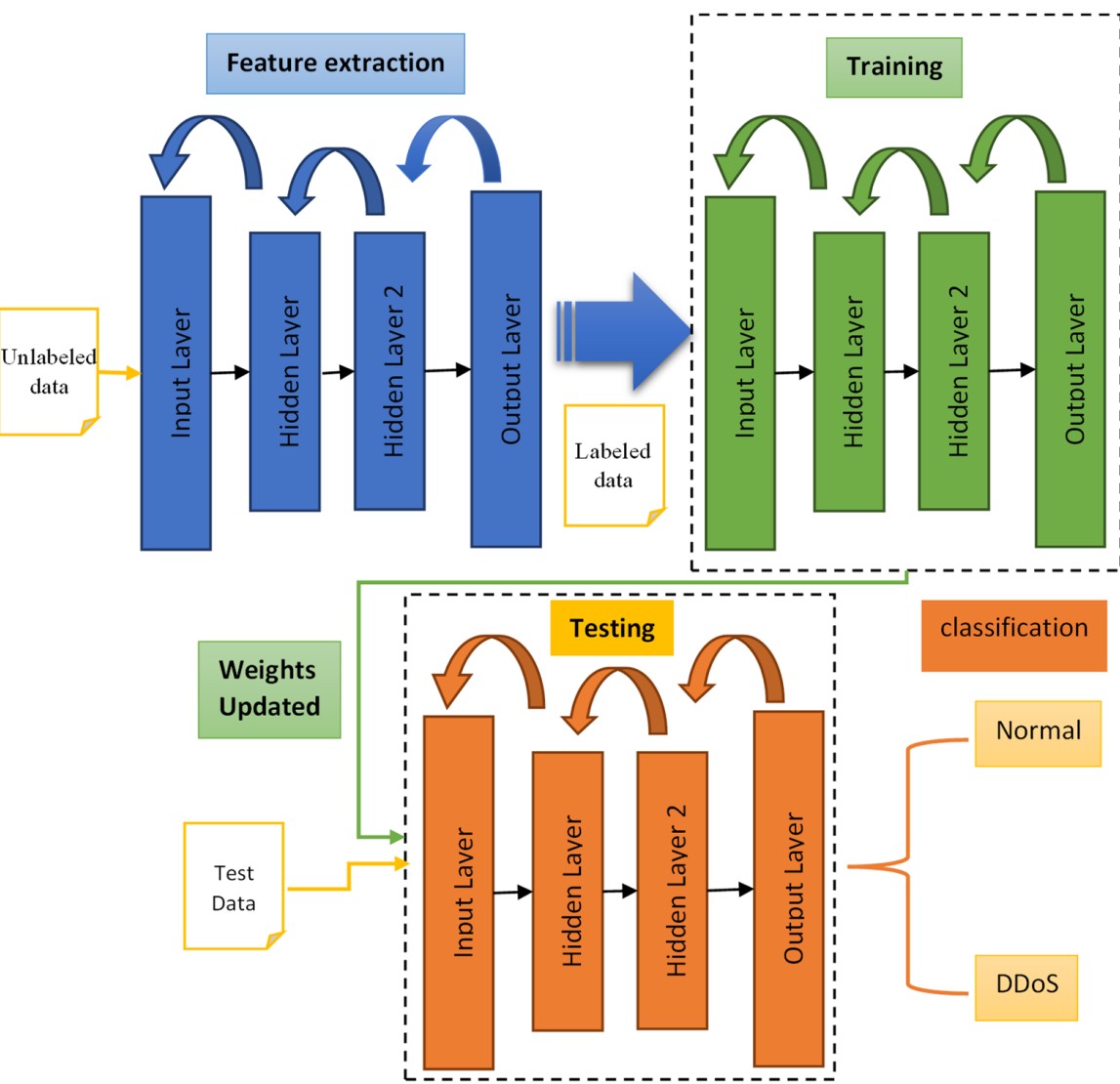

**Figure 1.** Proposed deep-learning model with autoencoder structure.

In supervised learning, the input-to-output relationship is modeled in two phases: Training and examination. In training, the classification model is constructed by analyzing the training dataset during the learning phase, i.e., the training process. Using the 'TCP-dump' networking tool, data in the form of 'pcap' files are captured in real-time. This allows for labels to be added to network traces in real-time, indicating that these traces can then be used for training. During the testing phase, new instances are classified using

the model developed during the training phase. The mapping between active and output network traffic is determined using the supervised learning algorithm. The first obstacle to network traffic classification is obtaining a labeled dataset. Using a portion of the data as a training set (e.g., 80% for training) and the remainder as a testing set (e.g., 20% for testing) is one solution. The second obstacle is the possibility of newly generated network traffic to not belong to known traffic classes. The third obstacle is classifying traffic in real-time, i.e., during the online mode. Based on [29], the following flow instance data can be used to train a DL algorithm:

- Source IP, as well as destination IP along with port number;
- The protocol type (TCP, UDP, or ICMP) and header length;
- The number of packets transmitted at every switch;
- The number of packets received at each switch;
- The packet count (the number of packets within each flow).

In SDN infrastructure, the controller is an agent. The controller monitors the network status to make decisions regarding data forwarding:

- Feature extraction through normalization and autoencoder;
- Training the model using deep neural network;
- Classifying the traffic for one of the two classes: Normal and DDoS.

An autoencoder is a feedforward neural network that has one or more hidden layers. It is a type of unsupervised neural network, where the network attempts to match outputs to input vectors as closely as possible. Additionally, it can be used to generate higher or lower dimensionality representation of inputted data. The use of unsupervised learning of compressed data encoding makes neural networks extraordinarily versatile. In addition, these networks can be trained one layer at a time, which minimizes the computational resources needed to design an effective model [30]. If the hidden layers are less dimensional than the input and output layers (as shown in Figure 2), then the network will be used for data encoding (as it allows for compression). Multilayered autoencoders can be trained in series, allowing for the gradual compression of information, creating what is called a stacked autoencoder [31].

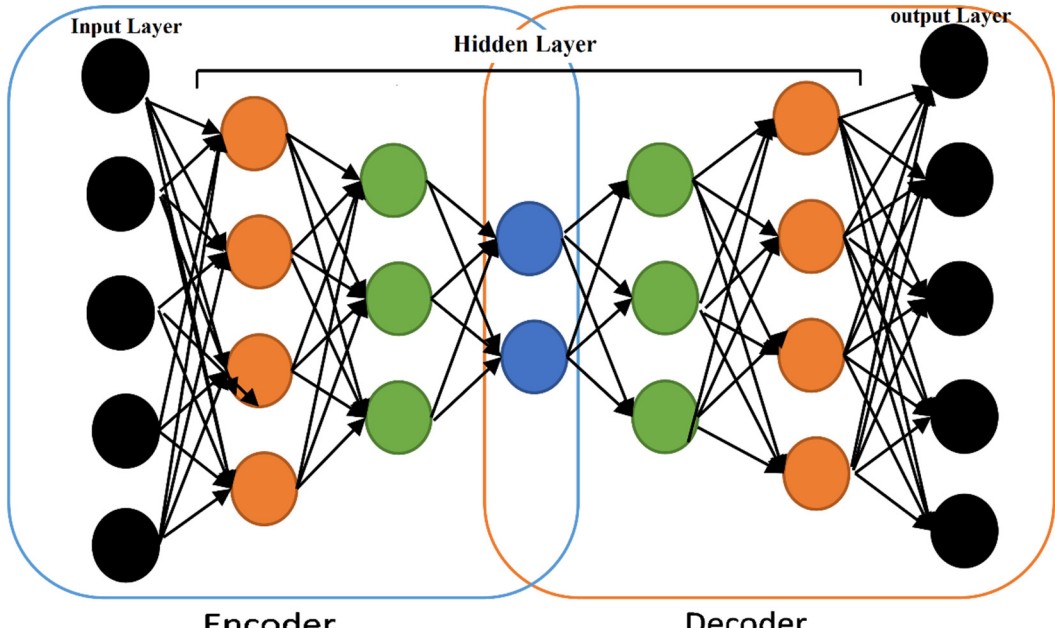

**Figure 2.** Deep autoencoder.

The self-encoding model consists of input, hidden, and output layers. The flow table feature vector is [32]:

$$x_i = \left[ x_{i1}, x_{i2}, x_{i3}, \ldots, x_{ij} \right]^T$$

where $i$ represents the $i$-th flow table feature vector, and $j$ represents each flow table feature. The vector contains $j$ features. The hidden layer encodes and compresses the input feature vector of the flow table according to Equation (1) [33]:

$$encoder = W_1 x_i + b_1 \tag{1}$$

where $W_1$ is the weight connecting the input layer and the hidden layer, $x_i$ is the input feature vector of the $i$-th flow table, and $b_1$ is the bias of the hidden layer neuron.

After the encoding is completed and determined on the output result of the hidden layer, the output layer is decoded and reconstructed to produce an output of the same size as the input layer neuron, using Equation (2) [33]:

$$decoder = f(W_2(encoder)_i + b_2) \tag{2}$$

where $f$ is the activation function, $W_2$ is the weight between the hidden and the output layer, $(encoder)_i$ is the stream table feature vector compressed by the hidden layer coding, and $b_2$ is the bias of the output layer neuron.

Finally, the goal of training the self-encoding model is achieved by minimizing the loss function using Equation (3) [34]:

$$loss = \sum_{i=1}^{n} (x_i - (decoder_i))^2 \tag{3}$$

where $n$ is the number of flow table feature vectors, $x_i$ is the input flow table feature vector, and $(decoder_i)$ is the flow table feature vector output by $x_i$ through the self-encoding model.

To achieve dimensionality reduction and feature extraction when constructing the model, we intend to use the deep stack auto-encoding model [35]. The deep stack auto-encoding model is constructed by stacking the input layer and hidden layer of self-encoding models' layer by layer. Each self-encoding model generates a hidden layer. After the flow table feature vector is learned by the first self-encoding model, the compressed abstract features are obtained in its hidden layer, and the hidden layer of the first self-encoding model is used as the input layer of the second self-encoding model. The learning of the second auto-encoding model indicates that more abstract features are obtained after further compression in its hidden layer. Then, the text of the second auto-encoding model is used to achieve the purpose of dimensionality reduction and abstract feature extraction when constructing the model.

When building a deep-learning model, the use of convolutional layers of different depths will have a significant impact on the detection accuracy of the model, and training the model leads to better performance. Two models have been investigated, one based on multilayer convolutional neutral networks (CNN) with Max pooling, and the second based on bidirectional long short-term memory (BDLSTM). We intend to use a batch size of 50 for model training by default. We will test CNN models containing three convolutional layers, two max pooling layers, one flatten, and two dense layers (as described in Table 1).

The BDLSTM model has one LSTM and four dense layers (the structure is provided in Table 2).

**Table 1.** CNN structure.

| Layer (Type) | Output Shape |
|---|---|
| Conv 2D | (None, 48, 48, 50) |
| Max Pooling | (None, 24, 24, 50) |
| Conv 2D | (None, 22, 22, 64) |
| Max Pooling | (None, 11, 11, 64) |
| Conv 2D | (None, 9, 9, 64) |
| flatten (Flatten) | (None, 5184) |
| Dense | (None, 64) |
| Dense | (None, 1) |

**Table 2.** BDLSTM Structure.

| Layer (Type) | Output Shape |
|---|---|
| Dense | (None, 18, 64) |
| Bidirectional | (None, 128) |
| Dense | (None, 32) |
| Dense | (None, 16) |
| Dense | (None, 12) |
| Dense | (None, 1) |

## 5. Evaluation Metrics

Metrics commonly used for evaluation include training accuracy, validation accuracy, recall, precision, F1-score, and the confusion matrix. These metrics are calculated using the following equations [36,37]:

1.  Training/validation accuracy: This metric measures the percentage of true detections through total traffic trace. It is computed as follows:

$$\text{Trained/validate Accuracy} = \frac{TP + TN}{TP + TN + FP + FN} \tag{4}$$

where *TP* is the (true positive), which is the number of anomaly records that is correctly classified. *TN* is the (true negative), which represents the number of normal records that is correctly classified. *FP* is the (false positive), which is the number of normal records that is incorrectly classified. *FN* is the (true negative), which represents the number of anomaly records that is incorrectly classified.

2.  Recall: This metric is used to show the percentage of predicted intrusions against all intrusions presented. The aim is to achieve higher recall values. It is computed using the following equation:

$$\text{Recall} = \frac{TP}{TP + FN} \tag{5}$$

3.  Precision: This metric is used to show the many intrusions predicted by the intrusion detection system (NIDS), which are actual intrusions. The aim is to achieve higher precisions than the lower false alarms. It is computed using the following equation:

$$\text{Precision} = \frac{TP}{TP + FP} \tag{6}$$

4.  F1-score: This metric attempts to better measure the accuracy of an intrusion detection system (NIDS) by considering both the precision and recall. The aim is to achieve higher F1-scores. It is computed as follows:

$$\text{F1-Score} = \frac{2 \times Precision \times Recall}{Precision + Recall} \tag{7}$$

Using the confusion matrix (CM), all of the above metrics can be obtained, as well as receiver operating characteristics (ROC).

## 6. Results and Discussion

The model was tested using the ISCXIDS2012 dataset, and then using simulation topology for the generation of normal and DDoS attacks.

### 6.1. Test Using ISCXIDS2012

The model was tested to observe whether it could detect DDoS attacks from the ISCXIDS2012 dataset. The input parameters are 50,000 for regular traffic and 50,000 for DDoS attacks, in which each involves a 50,000 flow status interval. The data were split into 60% for training, 20% for validation, and 20% for testing. Attacks were labeled with 1 and 0 for the regular traffic, and the model was trained for 20 epochs (Table 3; Figure 3).

**Table 3.** Results of tests using the ISCXIDS2012 dataset for three models.

| Network | Loss | Accuracy | Val. Loss | Val. Accuracy |
|---|---|---|---|---|
| ANN-Autoencoder | 0.5842 | 0.6612 | 0.5641 | 0.6484 |
| CNN-Autoencoder | 0.1027 | 0.9554 | 0.5907 | 0.6279 |
| BDLSTM-Autoencoder | 0.0388 | 0.9935 | 0.0624 | 0.9930 |

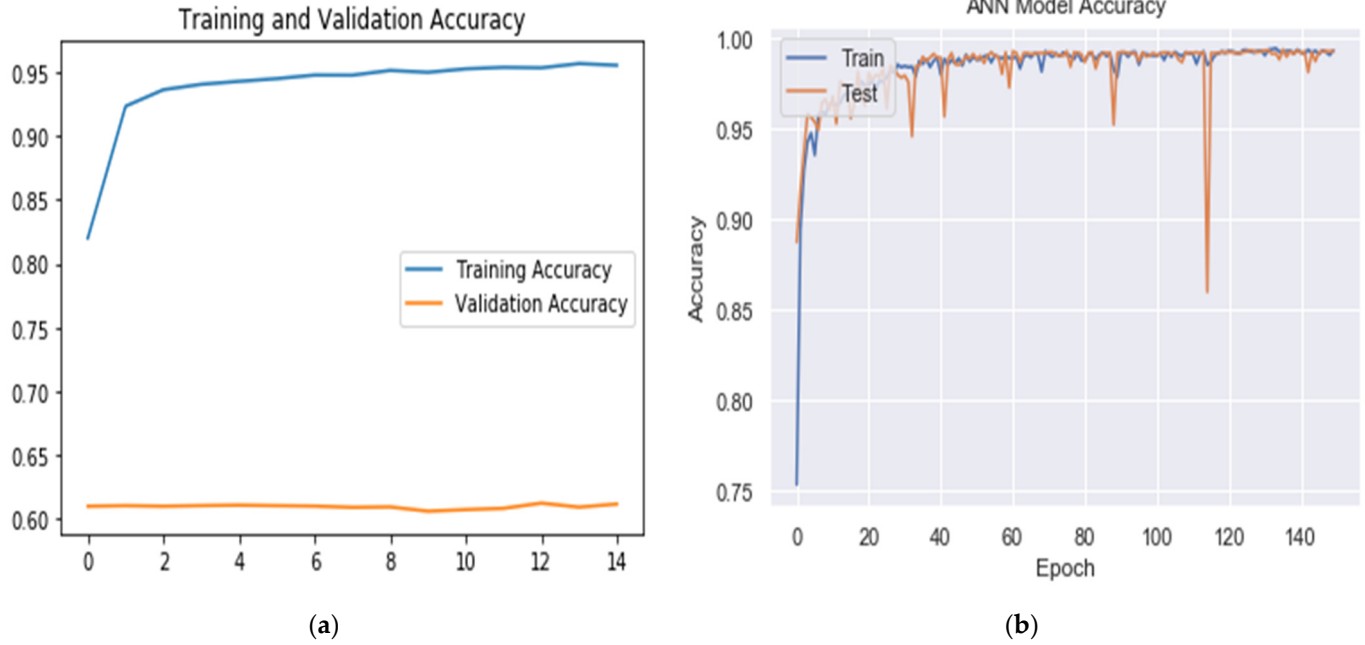

(**a**)    (**b**)

**Figure 3.** Accuracy results for (**a**) the CNN-autoencoder model, and (**b**) the BDLSTM-autoencoder model.

The single autoencoder model was less accurate in both training and validation, achieving 64.05% for training and 62.11% for validation. The CNN-autoencoder model suffered from overshooting, in which the training accuracy reached a high of 95.54%, while the validation accuracy was around 61.14%. The BDLSTM-autoencoder model achieved the highest result with a high of 99.35% in training and 99.30% in validation, in which the two are very close. Based on these results, we selected the BDLSTM-autoencoder model as the primary DDoS classifier, subject to more testing in the future.

### 6.1.1. Effect of Data Splitting

The effect of splitting on DDoS detection was investigated using train-test-validate split in three different ways: (60-20-20), (70-15-15), and (80-10-10), as shown in Table 4.

**Table 4.** DDoS detection results under the BDLSTM-autoencoder model for three different train-test-validate data splits.

| Network | Accuracy | Val. Accuracy | Precision | Recall | F1-Score |
|---|---|---|---|---|---|
| (60, 20, 20) splitting | 0.9935 | 0.9930 | 0.99 N<br>0.99 At | 0.99 N<br>0.99 At | 0.99 N<br>0.99 At |
| (70, 15, 15) splitting | 0.9875 | 0.9826 | 0.98 N<br>0.99 At | 0.99 N<br>0.98 At | 0.99 N<br>0.99 At |
| (80, 10, 10) splitting | 0.9927 | 0.9884 | 0.97 N<br>1.0 At | 1.0 N<br>0.97 At | 0.99 N<br>0.99 At |

N: Normal traffic; At: DDoS Attack.

In Table 4 and Figure 4a–f, the tests found that splitting was not very influential on the results, as no significant differences are found in accuracy and other metrics. However, 60-20-20 splitting achieved relatively more accuracy and a more stable result.

6.1.2. Effect of Activation Function in Output Layer Neurons

The effect of using different activation functions in output layer neurons of the BDLSTM-autoencoder model on DDoS detection was investigated using sigmoid ReLU or SoftMax (Table 5).

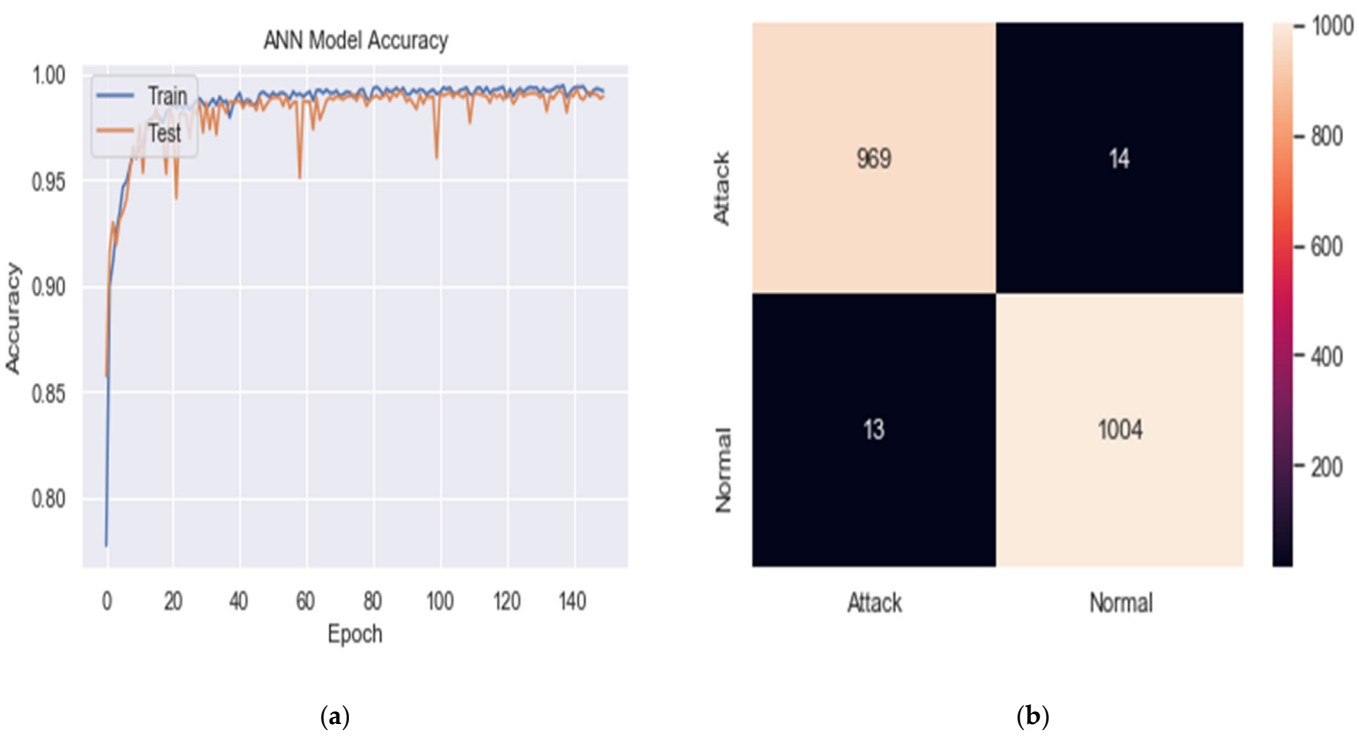

(**a**)                                                                                   (**b**)

**Figure 4.** *Cont*.

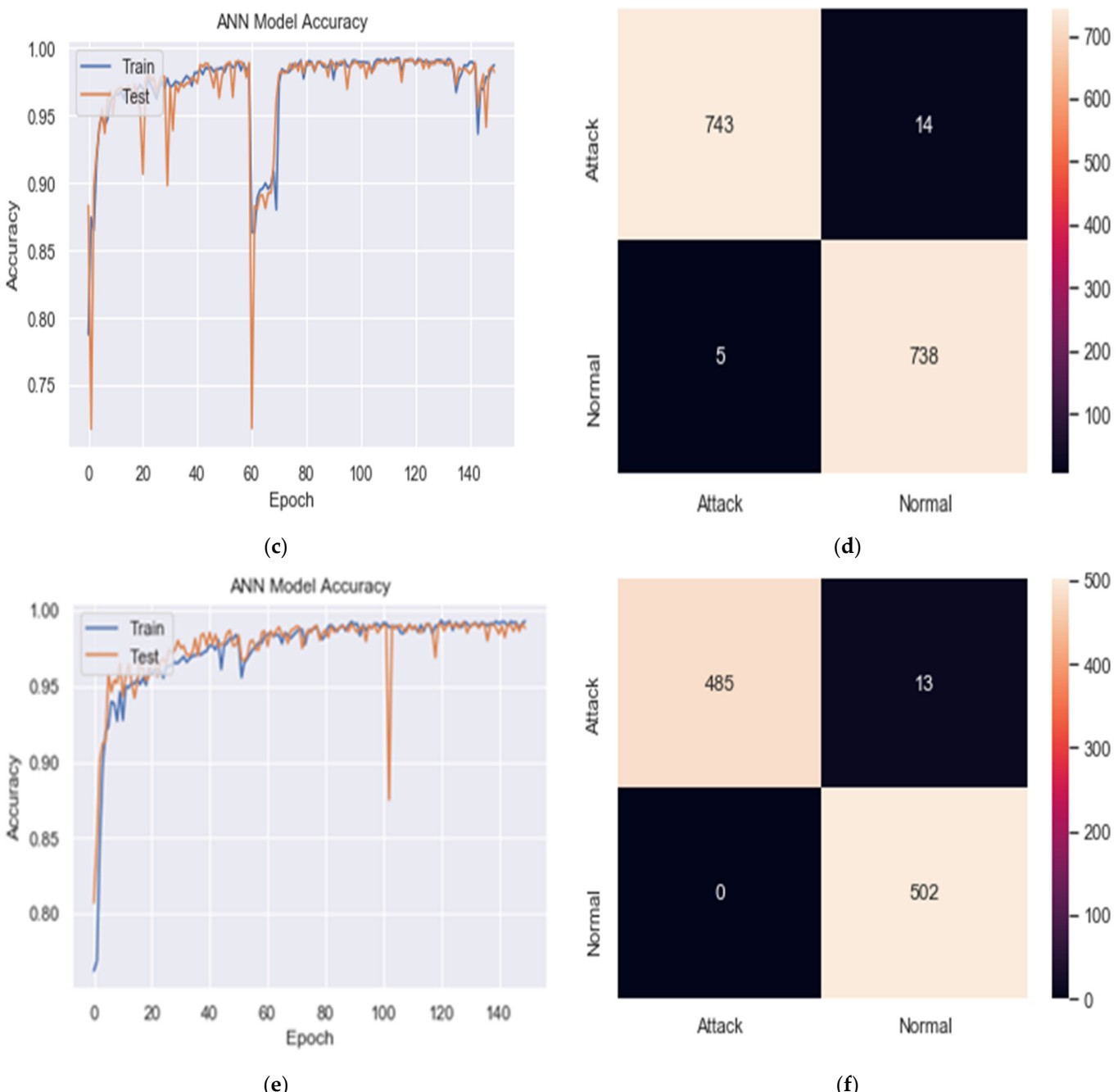

**Figure 4.** Training and validation accuracy and confusion matrix of DDoS detection results under the BDLSTM-autoencoder model. (**a**) Training and validation accuracy for (60, 20, 20) splitting, (**b**) confusion matrix for (60, 20, 20) splitting. (**c**) Training and validation accuracy for (70, 15, 15) splitting, (**d**) confusion matrix for (70, 15, 15) splitting. (**e**) Training and validation accuracy for (80, 10, 10) splitting, (**f**) confusion matrix for (80, 10, 10) splitting.

**Table 5.** BDLSTM-autoencoder model DDoS detection results for four activation function models.

| Network | Accuracy | Val. Accuracy | Precision | Recall | F1-Score |
|---|---|---|---|---|---|
| Proposed model | 0.9935 | 0.9930 | 0.99 N<br>0.99 At | 0.99 N<br>0.99 At | 0.99 N<br>0.99 At |
| ReLU | 0.9554 | 0.6114 | 0.00 N<br>0.49 At | 0.00 N<br>1.00 At | 0.00 N<br>0.65 At |
| SoftMax | 0.9935 | 0.9930 | 0.00 N<br>0.48 At | 0.00 N<br>1.00 At | 0.00 N<br>0.65 At |
| tanh | 0.4920 | 0.5148 | 0.51 N<br>0.00 At | 1.00 N<br>0.00 At | 0.68 N<br>0.00 At |

N: Normal traffic; At: DDoS Attack.

As shown in Figure 5 and Table 5, the experiments found that when the output layer neurons use the sigmoid activation function, detection accuracy is better than when using activation functions, such as ReLU, SoftMax, or tanh. The sigmoid function better solves the linear bottleneck problem, and the resulting model is easier to train.

### 6.2. Test Using UNSW2018

The model was tested to observe whether it could detect DDoS attacks from the UNSW2018 dataset [17]. The input parameters comprise 100,000 for regular traffic and 100,000 for DDoS attacks, in which each involves a 100,000 flow status interval. The data were split into 80% for training and 20% for testing. Attacks were labeled with 1 and 0 for the regular traffic, and the model was trained for 10 epochs (Table 6; Figure 6).

Similar to the previous dataset, the single autoencoder model was less accurate in both training and validation, achieving 67.02% for training and 62.11% for validation. The CNN-autoencoder model suffered from overshooting, in which the training accuracy reached a high of 95.54%, while the validation accuracy was around 61.14%. The BDLSTM-autoencoder model achieved the highest result, reaching a high of 99.95% in training and 99.94% in validation, in which the two are very close. Based on these results, we selected the BDLSTM-autoencoder model as the primary DDoS classifier, subject to more testing in the future.

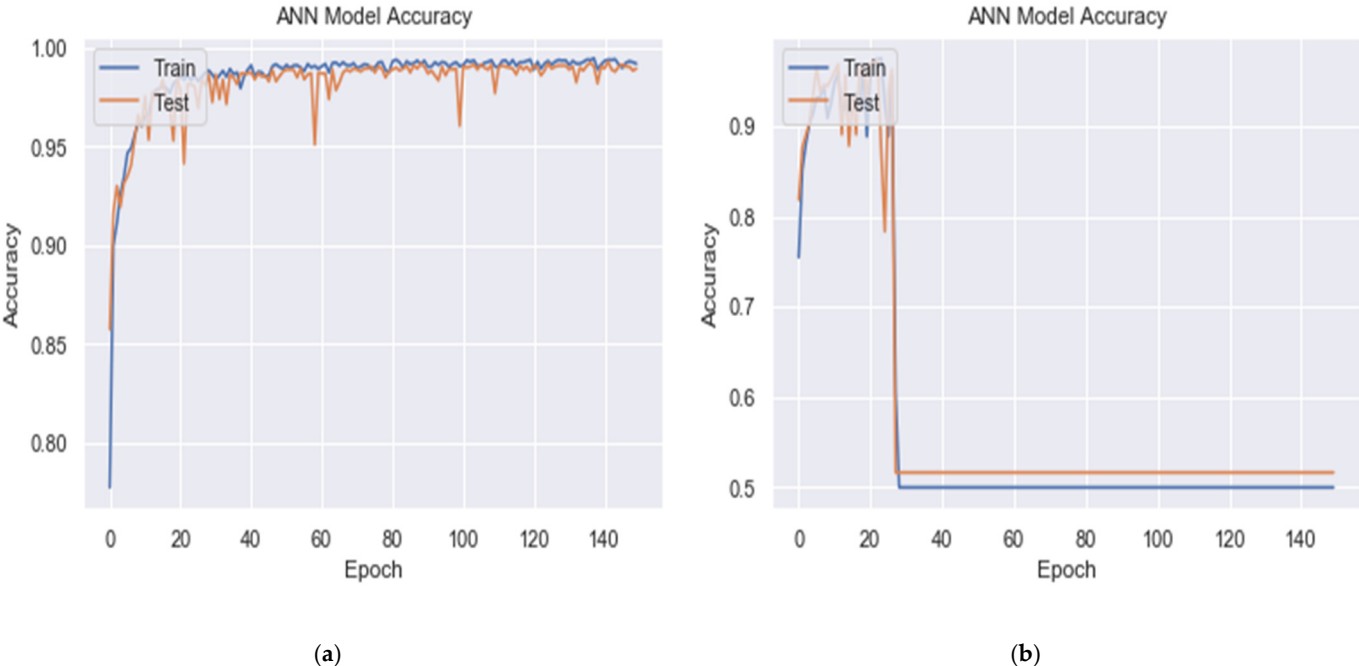

(**a**)                    (**b**)

**Figure 5.** *Cont*.

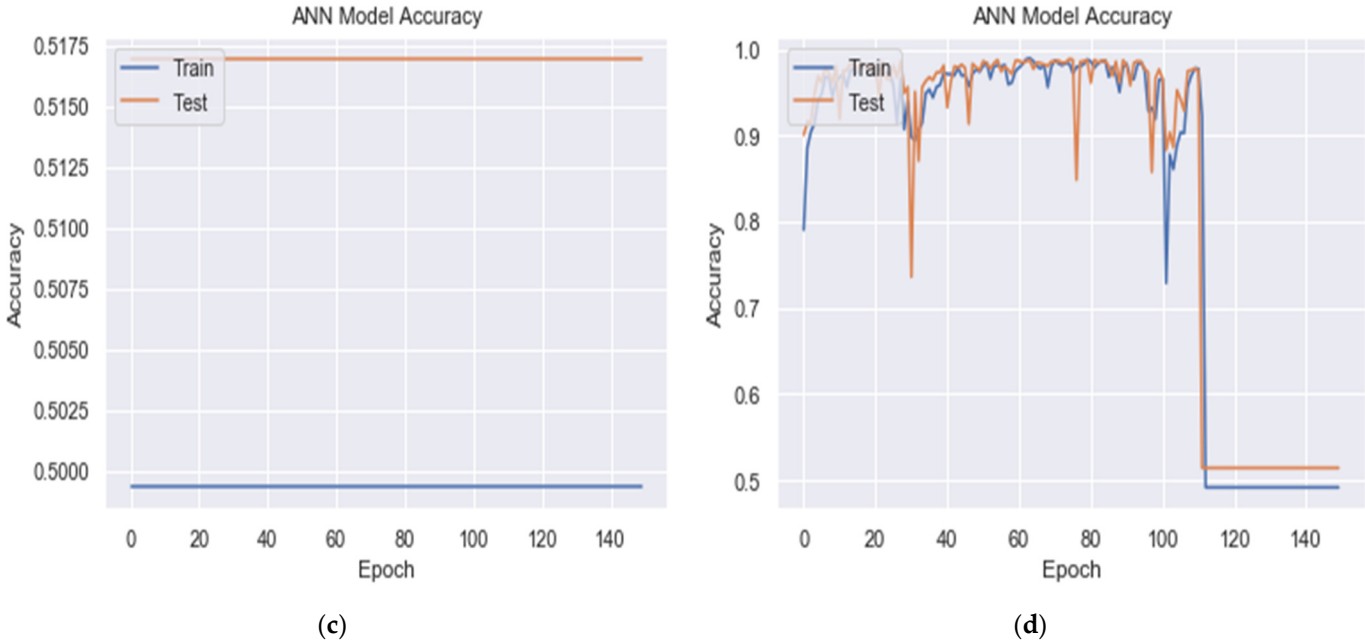

(**c**)                (**d**)

**Figure 5.** Training and validation accuracy for DDoS detection of the BDLSTM-autoencoder model using different activation functions at output layer (**a**) sigmoid, (**b**) RelU, (**c**) SoftMax, and (**d**) tanh for 80-10-10 splitting.

**Table 6.** Results of tests using the UNSW2018 dataset for three models.

| Network | Loss | Accuracy | Val. Loss | Val. Accuracy |
|---|---|---|---|---|
| ANN-Autoencoder | 0.5746 | 0.6453 | 0.5787 | 0.6512 |
| CNN-Autoencoder | 0.1338 | 0.9611 | 0.0880 | 0.986 |
| BDLSTM-Autoencoder | 0.0020 | 0.9995 | $4.9197 \times 10^{-4}$ | 0.9994 |

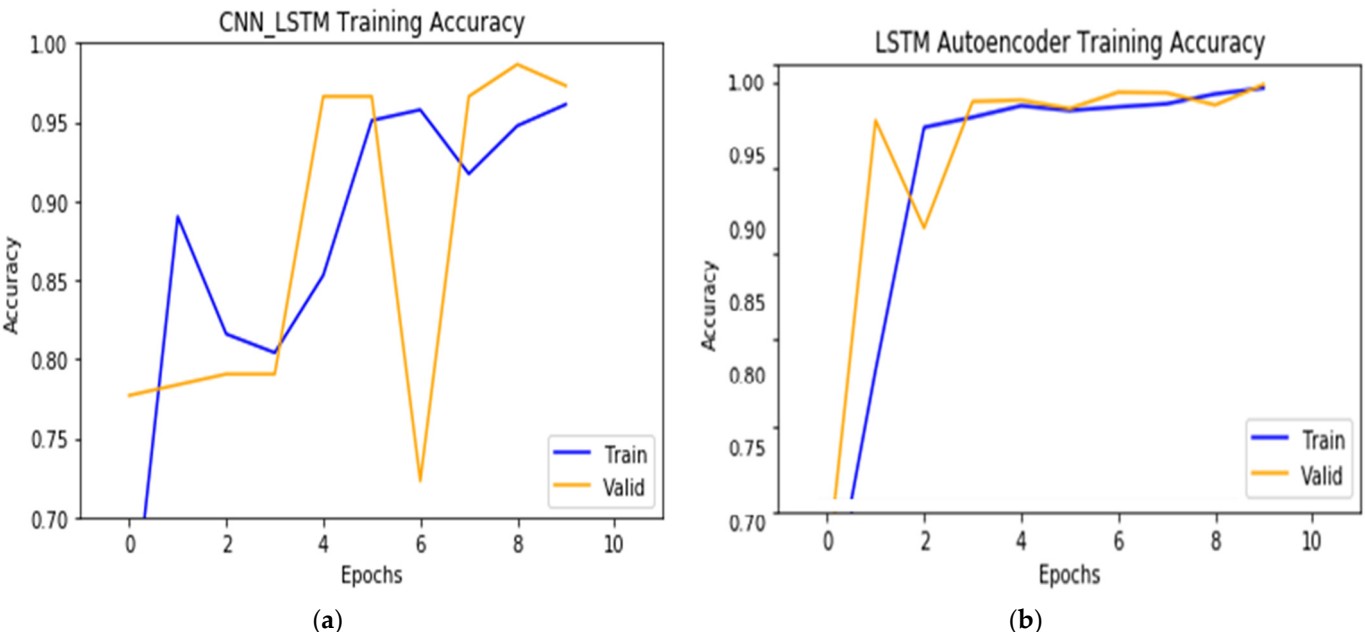

(**a**)                (**b**)

**Figure 6.** Accuracy results for (**a**) the CNN-autoencoder model, and (**b**) the BDLSTM-autoencoder model.

### 6.3. Comparison with Previous Work

The result of the proposed system was compared with some recent related works using the ISCXIDS2012 dataset and UNSW2018 BoTIoT (Table 7).

**Table 7.** Comparison results between BDLSTM-autoencoder model and some recent works.

| Ref | Dataset | Algorithm | Accuracy | Val. Accuracy | Precision | Recall | F1-Score |
|---|---|---|---|---|---|---|---|
| Proposed model | ISCXIDS2012 | BDLSTM-Autoencoder | 0.9935 | 0.9930 | 0.9978 N 0.9991 At | 0.99 N 0.99 At | 0.9981 N 0.9987 At |
| Dehkordi et al. [38] | | Model | 0. 8711 | —— | 0. 3708 N 0. 3574 At | — | 0.4580 N 0.5266 At |
| | | Naive Bayes | 0.9584 | —— | 0.9156 | —— | 0.9116 |
| | | Random Tree | 0.9984 | —— | 0.9966 | —— | 0.9967 |
| Proposed model | UNSW2018 | BDLSTM-Autoencoder | 0.9995 | 0.9994 | 0.95 N 0.99 At | 0.94 N 0.99 At | 0.95 N 0.99 At |
| Ivanova et al. [19] | | Model | 0.9999 | 0.9999 | 0.8255 N 0.9999 At | 0.6635 N 0.9999 At | 0.7357 N 0.9987 At |
| Prasad et al. [20] | | Model | 0.9999 | 0.9999 | 0.8772 N 0.9999 At | 0.8255 N 0.9999 At | 0.8197 N 0.9999 At |

### 6.4. Test Using Dynamic Value

The OpenDaylight controller and Mininet emulator, which have been applied in the adopted work, were performed on a PC with 16 GB RAM and an Intel Core i7 processor. The Mininet emulator further tested the BDLSTM-autoencoder model's ability to detect DDoS attacks. A Scapy script inside Mininet generates UDP packets and spoofs the source IP address of the packets. The protocol configuration was DP:0, TCP:2, ICMP:3. The input parameters are shown in Table 8.

**Table 8.** Input parameters for network traffic implementation of an emulated test of the BDLSTM-autoencoder model's ability to detect DDoS attacks.

| Switch | Src | Dst | Pktcount | Bytecount | Dur | Dur_Nsec | Tot_Dur | Flows | ⋮ | Pktrate | Pairflow | Protocol | Port_No | Tx_Bytes | Rx_Bytes | Tx_Kbps |
|---|---|---|---|---|---|---|---|---|---|---|---|---|---|---|---|---|
| 7 | 10.0.0.3 | 10.0.0.10 | 247 | 24,206 | 535 | 41,000,000 | $2.53 \times 10^{11}$ | 13 | … | 0 | 1 | ICMP | 2 | 35,897 | 31,370 | 0 |
| 7 | 10.0.0.12 | 10.0.0.17 | 122,751 | 7,119,558 | 410 | 808,000,000 | $4.11 \times 10^{11}$ | 3 | … | 251 | 1 | TCP | 2 | 33,018,521 | 470,020,975 | 0 |
| 5 | 10.0.0.16 | 10.0.0.3 | 168,663 | 91,078,202 | 322 | 297,000,000 | $3.22 \times 10^{11}$ | 5 | … | 545 | 1 | TCP | 1 | 6,115,457 | 144,666,612 | 0 |
| 6 | 10.0.0.12 | 10.0.0.7 | 605 | 59,290 | 620 | 214,000,000 | $6.20 \times 10^{11}$ | 3 | … | 0 | 1 | ICMP | 3 | 65,744 | 135,525,618 | 0 |
| 4 | 10.0.0.2 | 10.0.0.8 | 35,970 | 38,344,020 | 78 | 820,000,000 | $7.882 \times 10^{11}$ | 6 | … | 451 | 0 | UDP | 3 | 3236 | 3404 | 0 |
| 4 | 10.0.0.9 | 10.0.0.2 | 792 | 77,616 | 811 | 590,000,000 | $8.12 \times 10^{11}$ | 5 | … | 0 | 0 | ICMP | 2 | 105,851 | 135,561,984 | 0 |

DeepInsight has been used to transform the data to a matrix format for CNN architecture.

Figure 7 shows converting non-image dataset to image dataset using the DeepInsight methodology, the feature density matrix, and DeepInsight for train data shown in Figure 7a,b. The green line consists of all feature data, while the red line represents the extracted data in DeepInsight. Moreover, the blue dots feature denotes extracted data in the density matrix.

The classification result is shown in Table 9, Figures 8 and 9.

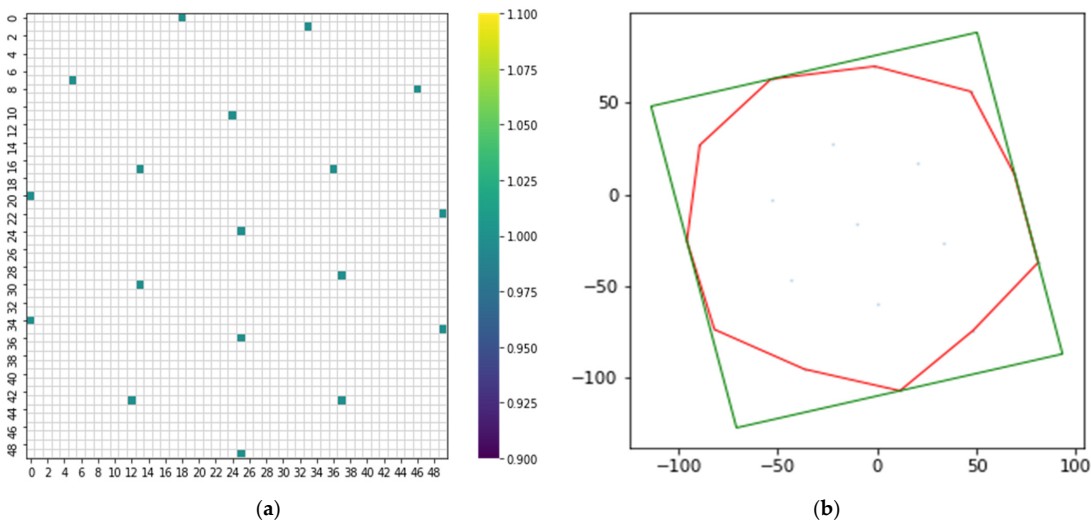

**Figure 7.** (**a**) Feature density matrix. (**b**) DeepInsight for train data.

**Table 9.** BDLSTM-autoencoder model results for DDoS detection of an emulated test of the BDLSTM-autoencoder model.

| Network | Accuracy | Val. Accuracy | Precision | Recall | F1-Score |
|---|---|---|---|---|---|
| Proposed model | 0.9762 | 0.9768 | 0.98 N<br>0.88 At | 0.92 N<br>0.97 At | 0.95 N<br>0.93 At |

```
Epoch 9/10
13846/13846 [==============================] - 9s 621us/step - loss: 0.0646 - accuracy: 0.9744
Epoch 10/10
13846/13846 [==============================] - 9s 618us/step - loss: 0.0625 - accuracy: 0.9756
6923/6923 [==============================] - 4s 525us/step - loss: 0.0602 - accuracy: 0.9762
Standardized: 97.68% (0.13%)
```

**Figure 8.** Screen shot for training and validation accuracy results from terminal output.

The system was quite capable of detection of DDoS attacks from dynamic data, reaching a high of 97.68% in accuracy.

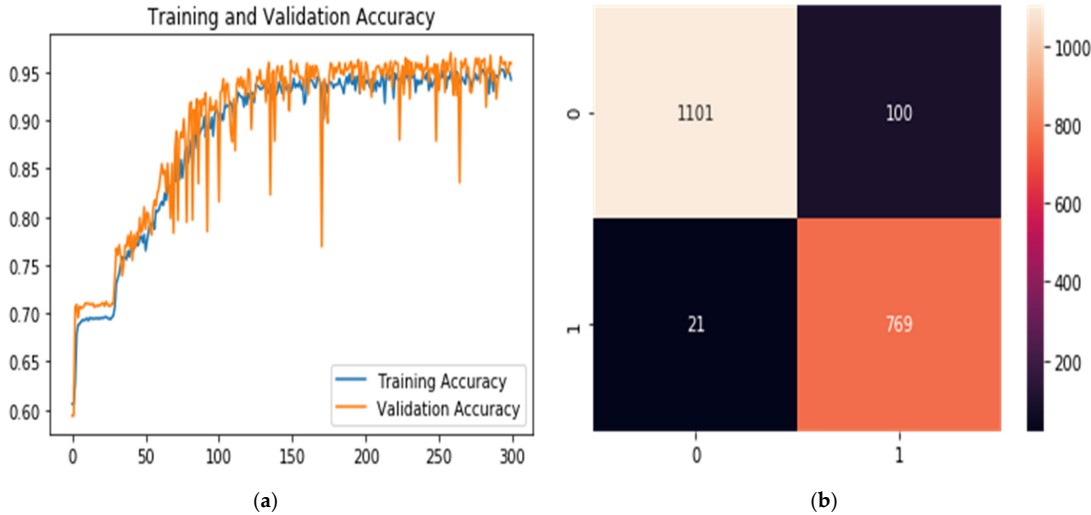

**Figure 9.** (**a**) Classification for training and validation accuracy results and (**b**) Confusion matrix.

## 7. Conclusions

In this work, the BDLSTM-autoencoder model, which combines deep neural network techniques with autoencoder-based feedforward neural networks, proved successful in terms of detecting DDoS attacks in an SDN environment. The deep stack auto-encoding model is constructed by stacking the input layer and hidden layer of self-encoding models' layer by layer. Each self-encoding model generates a hidden layer. After the flow table feature vector is learned by the first self-encoding model, the compressed abstract features are obtained in its hidden layer. Then, the hidden layer of the first self-encoding model becomes the input layer of the second self-encoding model. The learning of the second auto-encoding model obtains more abstract features after further compression in its hidden layer. Then, the auto-encoding model can be used to achieve the purpose of dimensionality reduction and abstract feature extraction when constructing the model. Two deep-learning models have been investigated, one based on multilayer convolutional neural networks (CNN) with Max pooling, and the second based on bidirectional long short-term memory (BDLSTM). In this article, two datasets that train and test DDoS attacks (ISCX-IDS-2012 and UNSW2018) were compared with related works. Additionally, the data generated make use of a Scapy script inside Mininet to create UDP packets and spoof the source IP address of the packets.

The model-based BDLSTM-autoencoder achieved higher accuracy than the CNN model. Dataset splitting had no significant effect on detection accuracy, although a 60-20-20 training, testing, and validation split was relatively better. The activation function in the output layer highly affected both the stability and accuracy of detection, with sigmoid as the best choice for model success. The ISCX-IDS-2012 dataset accuracy reached a high of 99.35% in training, 99.3% in validation, and 99.99% in precision, recall, and F1-score. In addition, the UNSW2018 dataset accuracy reached a high of 99.95% in training, 0.99.94% in validation, and 99.99% in precision, recall, and F1-score for attacks and 99.5%, 99.4%, and 99.5% in precision, recall, and F1-score, respectively. Moreover, the model achieved great results with a dynamic dataset (using an emulator), reaching a high of 97.68% in accuracy. However, further exploration on the use of autoencoders with other deep-learning techniques is still necessary, as well as testing with different DDoS datasets.

**Author Contributions:** Conceptualization, A.L.Y. and H.M.M.; methodology, A.L.Y. and H.M.M.; formal analysis, A.L.Y. and M.H.; investigation, A.L.Y. and H.M.M.; writing—original draft preparation, A.L.Y. and M.H.; supervision, H.M.M. and M.H. All authors have read and agreed to the published version of the manuscript.

**Funding:** This research received no external funding.

**Data Availability Statement:** Data are derived from public domain resources.

**Acknowledgments:** The authors would like to thank Computer Science Department, Faculty of Computers and Information, Menoufia University, Shebin Elkom, Egypt and Department of Information Systems, College of Administration and Economics, University of Baghdad, Iraq.

**Conflicts of Interest:** The authors declare no conflict of interest.

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
