# Peer review of "Improved DDoS Detection Utilizing Deep Neural Networks and Feedforward Neural Networks as Autoencoder"

_futureinternet, doi:10.3390/fi14080240_

Round 1
Reviewer 1 Report
The authors aimed to design autoencoder-based feedforward neural networks to detect DDoS attacks in an SDN environment. The proposed model was tested on ISCXIDS2012 dataset and an emulated data. However, there are quite a few comments for the authors to consider.
1. The proposed model was evaluated on only one public dataset. More experiments are expected on different datasets, as it is mentioned in related work.
2. The discussion of the previous work on DDoS attacks detection in the SDN environment is weak. More references of DDoS attacks detection for the SDN should be included.
3. The description of the emulator in the experiment is not clear. It is not sure if the proposed model can work in the real SDN environment.
4. The reason why the proposed model works for the SDN environment is not clear. How DDoS attacks detection of SDN is different from the others? What particular components are designed in the proposed model to detect the DDoS attacks in the SDN environment? It looks to me that the proposed model can work for the general DDoS attacks detection.
5. The comparison with other works is weak. Only traditional machine learning methods are compared, for example, navie bayes, random tree. More advanced deep learning models should be compared with the proposed model.
Reviewer 2 Report
Very interesting ant timely article. I think it deserves publication and I am recommending accept with minor corrections. But there are some minor issues that require your attention. I list these corrections below as feedback / comments, and I am looking forward to reading the updated version of this article.
- You have done a really good job at reviewing so many articles, but not a single article on IoT cyber risk and DDoS attacks (e.g., Mirai) use IoT devices. There are articles on the topic of IoT risk, that review recent and relevant literature, for example, on the related topic of ‘real-time intelligence for predictive cyber risk analytics’ - see: https://doi.org/10.1007/s42797-021-00025-1 and on the related topic of AI/ML dynamic cyber risk analytics at the edge - see: https://doi.org/10.1007/s42452-020-03559-4 - It would be interesting to see a few sentences reviewing and comparing your work in relations to these recent studies in related topics.
- One final comment, you should check if all the things discussed in the introduction, are also discussed in the conclusion. because the introduction is much longer than the conclusion. Usually, these sections are comparable in length. If you think you have covered everything, that’s OK, but just to mention that conclusion is the best chapter to outline your key findings and key conclusions. So, you should make use of this chapter to make your article more readable, and since most readers would focus a great deal of their attention on the conclusion, this section should make the key conclusions more visible (and hence more interesting).
Round 2
Reviewer 1 Report
The authors have revised the paper, by adding the explanation of the methods and experiment results. However, I suggest the authors further improve the presentation and polish the paper. For example, the figures in the experiment can be improved. In addition, what does "A" mean in Tables 4 and 5?
